# Factor Analysis of Subjective Well-Being Sustainability through Foreign Language Learning in Healthy Older Individuals

**Blanka Klimova** [1,*], **Marcel Pikhart** [1], **Szymon Dziuba** [2] **and Anna Cierniak-Emerych** [2]

[1]  Department of Applied Linguistics, Faculty of Informatics and Management, University of Hradec Kralove, 500 03 Hradec Kralove, Czech Republic; marcel.pikhart@uhk.cz
[2]  Department of Labour, Capital and Innovation, Faculty of Business and Management, Wroclaw University of Economics and Business, 53-345 Wrocław, Poland; szymon.dziuba@ue.wroc.pl (S.D.); anna.cierniak-emerych@ue.wroc.pl (A.C.-E.)
*   Correspondence: blanka.klimova@uhk.cz

**Abstract:** Healthy aging is one of the most important aspects of human life as it can significantly improve its quality. Therefore, it is necessary to promote successful aging as a significant and important part of maintaining physical and mental well-being in the elderly. One of the strategies to enhance the elderly's well-being may be also foreign language learning. The purpose of this study is to compare and discuss what effect foreign language learning (FLL) might have on subjective well-being among healthy older individuals in the Czech Republic and Poland, using factor analysis as the primary statistical method. The research sample consisted of two experimental groups of seniors; one from the Czech Republic (n = 92) and another from Poland (n = 100). The main research methods included a questionnaire survey and factor analysis. The factor analysis revealed the four significant factors and their correlations with demographic variables, whose results showed the effect of FLL on seniors' subjective well-being. In conclusion, learning a foreign language at an older age seems to be one of the key strategies to maintain a subjective feeling of happiness at a relatively high level in elderly people without necessary pharmacological intervention.

**Keywords:** foreign language learning; well-being; L2 acquisition; second language acquisition; psycholinguistics; language learning; societal sustainability; individual resilience; aging; healthy aging

## 1. Introduction

Currently, the number of older population groups is rapidly rising, especially in higher-income countries worldwide, and this demographic trend will probably continue in the next decades as well. Moreover, people at the age of 60+ years will form 35% of the total population in Europe by 2050 [1]. In addition, the majority of people in this age group suffer from mental or neurological disorders [2], which consequently impacts older people's performance of activities of daily living and worsens their quality of life.

Healthy aging is one of the most important aspects of human life as it can improve its quality significantly. Therefore, it is necessary to promote successful aging as a significant and important part of maintaining physical and mental well-being in the elderly [3]. As research [4–6] indicates, one of such nonpharmacological strategies enhancing the elderly's well-being (i.e., the state of being comfortable, healthy, or happy) may be also foreign language learning. Foreign language learning has not been traditionally considered to be an intervention method to improve well-being and quality of life. However, very recent research [4–6] proves that this activity can be very efficient in improving one's quality of life, particularly of the elderly. As Matsumoto [7] claims, foreign language learning is rewarding for seniors since it enriches their sense of meaning in their life. Furthermore, their motivation to learn a foreign language is high and especially intrinsic. They do not need any credits or diplomas to succeed; they simply study for the feeling of enjoyment and savoring and because they want to socialize with their peers [7,8].

Nevertheless, pleasant atmosphere and enhanced confidence may then also boost their cognitive gains [9], since research findings have proved that the aging brain is a dynamic set of biological features that can plastically reorganize against pathological decline [10]. In addition, research studies on bilingualism confirm that those knowing another language can delay the onset of dementia by several years in comparison with those who know only their native language [11–13]. This is a crucial finding as the epidemy of cognitive decline impairments in older age has recently gained its momentum due to the increased number of the elderly in our societies. To maintain our competitiveness and sustainability, it is crucial to look for various strategies that enable societal and individual resilience.

The purpose of this study is to compare and discuss what effect foreign language learning (FLL) might have on subjective well-being among healthy older individuals in the Czech Republic (CR) and Poland (PL) by using factor analysis as the primary statistical method. This research provides a systematic approach to the issues that are relevant and urgent as it can lead to societal sustainability and bring an alternative, i.e., nonpharmacological approach, to enhanced well-being in elderly people. As western society is aging very fast and the demographic trend towards elderly society is inevitable, it is crucial to create sustainable strategies towards societal resilience, and FLL can be one of the successful ones. Moreover, this research complements and develops research currently underway into healthy or successful aging.

## 2. Materials and Methods

### 2.1. Research Sample

The research sample consisted of two experimental groups of seniors; one from the Czech Republic (n = 92) and another from Poland (n = 100). Their age structure ranged from 55 years to 80+ years. All individuals were healthy seniors without any serious physical or mental impairments.

The age limit that was set for this survey was 55 years and older as this age is generally considered as the earliest retirement age. The upper limit was not set and some of the participants were older than 80 years. The age was not the most important parameter for the research, but it was only used as a reference point to have elderly respondents. The respondents were Czech or Polish citizens enrolled in the program of the University of the Third Age at the two universities based in the Czech Republic and Poland. All the respondents were enrolled in the language courses provided by these universities. All the courses were on the same, or very similar, language proficiency, according to the standardized and accepted Common European Framework Reference for Languages (CEFR) [14] corresponding to A2 to B1 level of language. The courses were held once a week for 90 min in two semesters corresponding to the academic year of the university, i.e., from October to December and from February to March. There was no significant change in the number of participants during the course, only a few individuals dropped out of the course for personal reasons.

### 2.2. Questionnaire Survey

The survey was conducted in May and June 2020 as an online questionnaire distributed to the participants of the survey via Google platform. The participants were notified to take part in the research by an email sent to them with a link to this questionnaire. All the participants of the research expressed their consent with the questionnaire by taking part in it; it was not obligatory, however, all the potential participants replied by answering the questionnaire.

The questionnaire consisted of a modified version of a well-known and established questionnaire widely used for testing satisfaction with language courses and other aspects of language education connected to non-language issues. The questionnaire was created by Woll and Wei [15], and this research utilized just the questions connected to subjective feelings of the well-being of the participants of the language courses. The questionnaire was also supplemented with further questions connected to the satisfaction with the course

and tutors; however, these questions were intended as filler questions and they did not have anything in connection with the research conducted. The questionnaire was created as concise as possible, taking into consideration the age of the participants and their possible attention span.

The questionnaire was entitled A satisfaction survey with the language classes so that the participants did not have any idea we were focusing on their subjective feelings of well-being. The surveys were conducted both in Czech and Polish, for Czech and Polish participants, respectively. It started with a short introduction and there were also a few questions regarding the age of the participants, the language studies, a town, city or village they have a permanent address in. The questionnaire itself contained 20 items to be answered on a standardized Likert Scale with six points as follows: Strongly agree, Agree, Agree a little, Disagree a little, Disagree, and Strongly disagree. The last question of the questionnaire was an open question about the satisfaction with the course; however, it again operated as a filler question and did not create any relevance for the research at all.

The questionnaire was available online through Google Forms, and all the data collected through this form were anonymous. There was no name, address or any other personal identifier collected. Each response only had a data stamp, fully anonymized. No IP address was collected either. All the contacted respondents answered the request and filled in the questionnaire. They reacted almost immediately, and the return of the questionnaires was very satisfactory, leading to high representability of the research as all the participants of the courses responded.

The questions focusing on the well-being were randomly mixed with those dealing with general satisfaction with the course; therefore, the respondents did not and could not imply that the research focused on their subjective feelings of well-being. The researchers were, however, not interested in their satisfaction with the course or the tutor. The only aim of this data collection was to obtain the data related to the subjective feeling of well-being, possibly connected to foreign language learning. This aspect was very important to eliminate the bias of the respondents if they had known the reason for this research. All the results collected were then analyzed by factor analysis.

### 2.3. Statistical Analysis

In the first stage, descriptive statistics and frequency distributions for demographic variables and those describing the level of foreign language learning were calculated. Next, average scores, deviations and medians were established for individual statements concerning the effect of language learning on well-being. Next, tests of intergroup differences were calculated by verifying whether the Czechs and the Poles differed in their responses to specific questions (Student's *t*-test for two independent groups or the Mann–Whitney U test if the variances in both groups differed significantly).

Then, factor analysis was performed to reduce the number of variables to several dimensions. The identified factors were used in the comparative analysis of the effect of foreign language learning on psychosocial well-being in Poland and the Czech Republic.

The significance level was set at $p < 0.05$, and the analysis was performed using the statistical software package PQ Stat 1.8.0 (PQ Stat Software, Poznan, Poland).

### 2.4. Research Questions

The research questions were formulated to obtain answers to the key ideas behind the research activity. They are as follows: First, what are the key factors affecting the seniors´ subjective well-being connected to foreign language learning? Second, is there any correlation between demographic variables and the key factors influencing well-being connected to foreign language learning? All these questions are answered in the results section and further developed in the context of other research in the discussion.

## 3. Results

### 3.1. Demographic Data

Figure 1 below illustrates the age structure and its distribution in both groups of respondents. The results show that most of the Poles (62%) were at the age between 55 and 60 years followed by the age group of 61–65 years. The age structure of the Czech seniors was more balanced, with the median being 66–70 years.

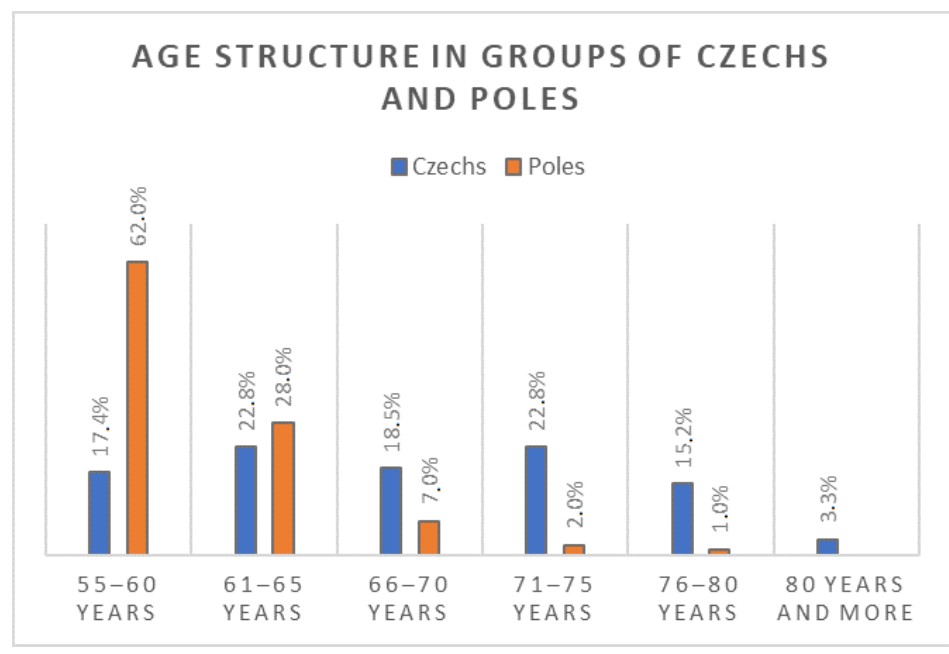

**Figure 1.** Age structure of the Czech and Polish seniors.

As for the sex, in the CR, the sample consisted of 35.9% males and 64.1% females, while in PL, the sample was comprised of 40% males and 60% females.

Regarding education, most of the Poles (92%) had university education, while only 58.7% of the Czech seniors had attained university education and 40.2% had attained secondary education. Only a few had primary education in both groups of the respondents (Figure 2, below).

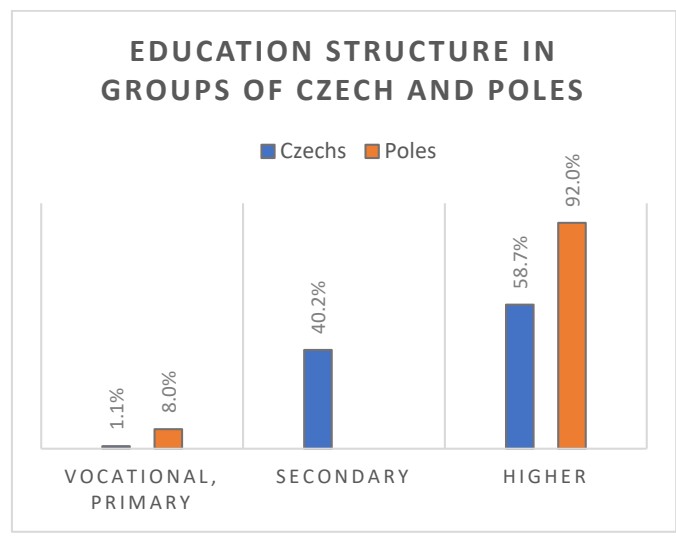

**Figure 2.** Education structure of the Czech and Polish seniors.

The data further reveal how many languages seniors were learning at the time of the survey. The results indicate that the majority of people in both groups were learning only one language (70% of seniors in PL and 67% of seniors in the CR).

### 3.2. The Effect of Language Learning on the Well-Being of Respondents: The Basic Approach

Table 1 below describes 20 statements on seniors' attitude towards foreign language learning. The statements in bold are the statements with which the respondents in both countries agreed most and those with which the respondents agreed least. The results also reveal that the Poles and the Czechs differed in their assessment of several positive and one negative aspect of language learning. The Poles appreciated each positive aspect more than the Czechs but also found foreign language learning more stressful (statement 15).

**Table 1.** Descriptive statistics for statements assessed on a scale of 1 to 6 (1-I absolutely disagree, 6-I absolutely agree).

| Statement | Country | Mean | Standard Deviation | Median | Significance of Differences (Student's *t*-Test/Mann–Whitney *U*-Test) |
|---|---|---|---|---|---|
| 1. Learning a new language improves my concentration. | CZ | 4.65 | 0.99 | 4 | $U = 2684.5$ |
| | PL | 5.38 | 0.76 | 6 | $p = 0.0001$ *** |
| 2. Learning a new language improves my memory. | CZ | 4.80 | 1.06 | 4 | $Uv = 3279$ |
| | PL | 5.36 | 0.77 | 5.5 | $p = 0.0002$ *** |
| 3. Learning a new language improves my attention. | CZ | 4.69 | 0.96 | 4 | $t = -3.764$ |
| | PL | 5.20 | 0.90 | 5 | $p = 0.0002$ *** |
| 4. Learning a new language improves my health. | CZ | 4.13 | 0.94 | 4 | $t = -1.667$ |
| | PL | 4.38 | 1.12 | 4 | $p = $ n.s. |
| 5. Learning a new language improves my creativity. | CZ | 4.42 | 0.99 | 4 | $t = -5.589$ |
| | PL | 5.16 | 0.84 | 5 | $p = 0.000001$ *** |
| 6. Learning a new language helps me find new friends. | CZ | 4.70 | 0.99 | 4 | $U = 3177$ |
| | PL | 5.22 | 0.76 | 5 | $p = 0.0001$ *** |
| 7. Learning a new language helps me understand different cultures. | CZ | 4.93 | 1.05 | 4 | $U = 3830$ |
| | PL | 5.27 | 0.75 | 5 | $p = 0.03$ * |
| 8. Learning a new language helps me while travelling. | CZ | 5.26 | 1.05 | 6 | $t = -1.695$ |
| | PL | 5.49 | 0.80 | 6 | $p = $ n.s. |
| 9. Learning a new language helps me with learning other things, too. | CZ | 4.49 | 0.92 | 4 | $t = -5.162$ |
| | PL | 5.13 | 0.80 | 5 | $p = 0.000001$ ** |
| 10. Learning a new language helps me while looking for life motivation. | CZ | 4.50 | 1.04 | 4 | $t = -1.138$ |
| | PL | 4.67 | 1.03 | 5 | $p = $ n.s. |
| 11. Learning a new language helps me with finding the purpose of my life. | CZ | 3.97 | 1.09 | 4 | $t = -2.869$ |
| | PL | 4.45 | 1.23 | 4 | $p = 0.004$ ** |
| 12. Learning a new language is enjoyable. | CZ | 4.84 | 1.09 | 4 | $t = -1.379$ |
| | PL | 5.05 | 1.05 | 5 | $p = $ n.s. |
| 13. Learning a new language brings me personal satisfaction. | CZ | 4.73 | 1.14 | 4 | $U = 3255.5$ |
| | PL | 5.31 | 0.79 | 5 | $p = 0.0001$ *** |
| 14. Learning a foreign language brings me feelings of happiness. | CZ | 4.72 | 1.13 | 4 | $t = -2.452$ |
| | PL | 5.10 | 1.03 | 5 | $p = 0.015$ * |
| 15. Learning a foreign language is stressful. | CZ | 2.43 | 1.23 | 2 | $t = -3.641$ |
| | PL | 3.13 | 1.40 | 3 | $p = 0.0003$ *** |
| 16. Learning a new language does not bring any benefits to me. | CZ | 1.91 | 0.91 | 2 | $t = 0.564$ |
| | PL | 1.83 | 1.11 | 1 | $p = $ n.s. |
| **17. Learning a new language can have a negative impact on me.** | **CZ** | **1.64** | **0.72** | **2** | $U = 4381.5$ |
| | **PL** | **1.82** | **1.08** | **2** | $p = $ n.s. |
| 18. Learning a new language occupies a lot of my time. | CZ | 2.93 | 2.93 | 3 | $t = -1.547$ |
| | PL | 3.21 | 1.27 | 3 | $p = $ n.s. |
| 19. Learning a new language is a positive motivation for me. | CZ | 4.63 | 1.11 | 4 | $t = -3.725$ |
| | PL | 5.18 | 0.94 | 5 | $p = 0.0002$ *** |
| 20. Learning a new language will be useful for me in the future. | CZ | 4.63 | 1.13 | 4 | $t = -3.617$ |
| | PL | 5.18 | 0.98 | 5 | $p = 0.0003$ *** |

* statistically significant difference at $p < 0.5$, ** statistically significant difference at $p < 0.01$, *** statistically significant difference at $p < 0.001$, n.s.-statistically not significant difference.

### 3.3. Factor Analysis

In order to reduce 20 statements to fewer categories, the factor analysis was performed and then new factors were used in the analysis on the effect of foreign language learning on seniors' well-being in comparative terms (Poland vs. the Czech Republic).

*(a) The assumptions of the factor analysis were verified.*
*1. Does the reduction of dimensionality make sense?*
The KMO coefficient (which compares partial correlations with the two-variable correlation coefficients) was calculated. KMO = 0, 9, 03. KMO > 0.5 means that reducing multidimensionality makes sense.
*2. Are the variables correlated with each other?*
Bartlett's test $\chi^2$ = 2349.82, $p < 0.001$
The variables are correlated with each other. Therefore, factor analysis for reducing the number of variables is justified.
*(b) An assumption was made about the number of isolated factors.*
If the number of variables is 20 or more, it is recommended to adopt the Kaiser criterion. This criterion assumes the inclusion of the components with eigenvalues >1.0 in the analysis. In this analysis, these will be 4 components (Table 2: eigenvalues).

**Table 2.** Eigenvalues and percentage of explained variance for the model with four factors.

| Factor | Eigenvalues | % Variance | Cumulative Eigenvalue | Cumulative % of Variance |
|---|---|---|---|---|
| 1 | 8.751592 | 43.75796 | | |
| 2 | 1.866923 | 9.334617 | 10.61852 | 53.09258 |
| 3 | 1.326967 | 6.634837 | 11.94548 | 59.72741 |
| 4 | 1.072815 | 5.364076 | 13.0183 | 65.09149 |

*(c) A cumulative percentage of the explained variance was calculated.*
For four factors, the model explains 65.1% of the variance (Table 2—cumulative percentage of variance).
*(d) The correlation matrix was analyzed and factors were identified.*
Four factors were identified based on factor loadings and factor content (Table 3, below).

**Table 3.** Factor loads.

| Variables | Factor Loadings | | | |
|---|---|---|---|---|
| | Factor 1 Positive Emotions and Mental Abilities (C1) | Factor 2 No Benefit (C2) | Factor 3 Future (C3) | Factor 4 Motivation(C4) |
| **Factor 1** | | | | |
| 1. Concentration | −0.725895 | | | |
| 2. Memory | −0.77291 | | | |
| 3. Attention | −0.775018 | | | |
| 12. Pleasure | −0.775248 | | | |
| 13. Satisfaction | −0.8336 | | | |
| 14. Happiness | −0.843787 | | | |
| **Factor 2** | | | | |
| 15. Stressful | | −0.50427 | | |
| 16. No benefit | | −0.513325 | | |
| 17. Negative effect | | −0.704568 | | |
| 18. Time-consuming | | −0.573559 | | |
| **Factor 3** | | | | |
| 7. Understanding of cultures | | | −0.484063 | |
| 20. Useful for the future | | | −0.428006 | |
| **Factor 4** | | | | |
| 10. Life motivation | | | | 0.352528 |
| 11. Life purpose | | | | 0.51503 |

*Factor 1*
Statements 2, 3, 12, 14, 19
2. Learning a new language improves my memory.

3. Learning a new language improves my attention.
12. Learning a new language is enjoyable.
14. Learning a foreign language brings me feelings of happiness.
19. Learning a new language is a positive motivation for me.
*Factor 2*
Statements 15, 16, 17, 18
15. Learning a foreign language is stressful.
16. Learning a new language does not bring any benefits to me.
17. Learning a new language can have a negative impact on me.
18. Learning a new language occupies a lot of my time.
*Factor 3*
Statements 7, 20
7. Learning a new language helps me understand different cultures.
20. Learning a new language will be useful for me in the future.
*Factor 4*
Statements 11, 10
10. Learning a new language helps me while looking for life motivation.
11. Learning a new language helps me with finding the purpose of my life.

In further analysis, the interpretation of <, > signs in the Tables below for factors 1, 2, 3 is reversed, because the factor loadings have negative values. The symbols of factors are C1, C2, C3, C4.

**(e) Individual factors were named.**
Factor 1: Positive emotions and mental abilities (C1)
Factor 2: No benefit (C2)
Factor 3: Future (C3)
Factor 4: Motivation (C4)

### 3.4. Analysis of Individual Demographic Variables and Characteristics of Language Learning

The analysis concerning **age** was conducted for the age groups that were studied in both countries: 55–60 years (denoted with code 1 in Table 4), 61–65 years (code 2), 66–70 years (code 3).

**Table 4.** Comparison of differences in the assessment of factors depending on age.

| Factor | Nationality | Test *H* | *p* | Differences between Age Groups |
|--------|-------------|----------|-----|-------------------------------|
| C1 | CZ | 12.93 | 0.001 * | 1 < 2. 2 < 3 |
| C1 | PL | 0.10 | 0.94 | |
| C2 | CZ | 5.35 | 0.069 | |
| C2 | PL | 0.01 | 0.995 | - |
| C3 | CZ | 9.00 | 0.011 ** | 1 < 3. 2 < 3 |
| C3 | PL | 4.83 | 0.089 | |
| C4 | CZ | 1.61 | 0.446 | |
| C4 | PL | 0.74 | 0.689 | |

Explanation: * $p < 0.01$, ** $p < 0.05$. Age group denotation: 1: 55–60 years; 2: 61–65 years; 3: 66–70 years.

The specific results of the analysis of the correlations between the perceived effects of foreign language learning on well-being depending on age were as follows: among the Czechs, the youngest group (55–60 years) significantly appreciates the factor positive emotions and mental abilities (C1) more than the group aged 61–65 years, while the group aged 61–65 years appreciates it significantly more than the group aged 66–70 years. In the assessment of the positive effect of languages on future life (C3), the groups aged 55–60 years and 61–65 years differ from the group aged 66–70 by significantly higher assessments. Consult Table 4 below.

Among Polish respondents, the analysis does not show any significant differences in the assessment of individual factors depending on age.

The analysis of the differences in the perceived effect of language learning on well-being between **the sexes** revealed the following statistically significant relationships:

Among the Czech seniors, men feel the negative effects of language learning less than women (C2), while Polish men appreciate pleasure and increased mental capacity less than women (C1). There are no significant differences between the sexes in the assessment of the other factors (Table 5).

**Table 5.** Comparison of differences in the assessment of factors depending on sex.

| Factor | Nationality | *U* Test | *p* | M vs. K |
|---|---|---|---|---|
| C1 | CZ | 4.46 | 0.108 | M < K |
|  | PL | 875.5 | 0.022 * | M > K |
| C2 | CZ | 640 | 0.006 * | M > K |
|  | PL | 1136.5 | 0.657 | M < K |
| C3 | CZ | 824 | 0.225 | M < K |
|  | PL | 1046.5 | 0.282 | M < K |
| C4 | CZ | 936 | 0.763 | M < K |
|  | PL | 1055.5 | 0.311 | M > K |

Explanation: * *p* < 0.05. Group denotation: M—men, K—women.

The relationship between the advantages of language learning and **education in both groups** (Czechs and Poles) is similar. People with a lower level of education are more likely to see no benefits than those with higher education (C2). Furthermore, people with a lower level of education are less likely to see the advantages of language learning for improving their future lives (Table 6).

**Table 6.** Comparison of differences in the assessment of factors depending on the level of education.

| Factor | Nationality | *U* Test | *p* | Secondary and Lower Education vs. Higher Education * |
|---|---|---|---|---|
| C1 | CZ | 950 | 0.807 | |
|  | PL | 962 | 0.769 | |
| C2 | CZ | 701 | 0.021 * | S < H |
|  | PL | 704 | 0.017 * | S < H |
| C3 | CZ | 710 | 0.026 * | S > H |
|  | PL | 711 | 0.020 * | S > H |
| C4 | CZ | 903 | 0.530 | |
|  | PL | 933 | 0.598 | |

Explanation: * *p* < 0.05. Group denotation: S—secondary and lower education, H—higher education. In the Czech group, people with lower levels of education than higher education are those with secondary education, whereas in the group of Poles—with primary or vocational education, people with higher education are those with bachelor's or master's degrees.

The analysis of the differences in perceived effects of foreign language learning on well-being between **the number of languages the respondents learnt** at the time of the survey revealed the following statistically significant relationships:

Among the Poles, learners of two languages and learners of three languages significantly appreciate the factor of positive emotions and mental abilities (C1) more than learners of one language, while no differences in this respect were found in the Czech group.

Furthermore, Czech learners of two languages have less appreciation of the positive effect of language learning on future life (C3) than learners of one or three languages. Consult Table 7 below.

**Table 7.** Comparison of differences in the assessment of factors depending on the number of foreign languages learnt by the respondent at the time of the survey.

| Factor | Nationality | Test *H* | *p* | Differences between Groups of Learners of Different Numbers of Languages |
|---|---|---|---|---|
| C1 | CZ | 4.457 | 0.107 | |
| | PL | 10.737 | 0.005 * | 2 < 1. 3 < 1 |
| C2 | CZ | 0.694 | 0.707 | |
| | PL | 0.523 | 0.770 | |
| C3 | CZ | 6.095 | 0.047 ** | 2 > 1. 2 > 3 |
| | PL | 1.034 | 0.596 | |
| C4 | CZ | 1.071 | 0.585 | |
| | PL | 0.917 | 0.632 | |

Explanation: * $p < 0.01$, ** $p < 0.05$. Group denotation: 1: learning 1 language, 2: learning 2 languages, 3: learning 3 languages. Learners of 3 and 4 languages were grouped into one group for the analysis.

## 4. Discussion

The results of the factor analysis revealed four significant factors associated with the effect of FLL on seniors' subjective well-being. These four factors are as follows: positive emotions and mental abilities (i.e., emotions and abilities evoked when learning a foreign language)—C1, no benefit (i.e., negative aspects of FLL)—C2, future (i.e., usefulness of FLL in future)—C3, and motivation (i.e., finding a new purpose of life)—C4. The findings in Table 1 clearly show that all positive factors prevail over the negative ones.

The respondents namely reported that FLL helped them in learning other things, improved their attention, concentration, and FLL especially helped them while travelling. In addition, the respondents on average disagreed with the statements covered under C2 (no benefit), such as FLL brings me no benefit, it is stressful or has a negative impact on me. Indeed, the respondents stated that FLL was enjoyable, brought them happiness, as well as they felt that their cognitive functions had improved [4,5,9,10,16–18]. This subjective feeling of improved cognitive functions is very important, regardless of the actual effect on the improvement of cognitive functions, as this subjective feeling improves the feeling of happiness and well-being significantly.

In general, there is not much research into the actual improvement of cognitive functions by foreign language learning; however, this improvement or at least maintenance of well-being is a sufficient nonpharmacological approach to successful aging. Generally, the respondents see FLL as a process rather than the aim of its own (cf. [6,9]); therefore, they do not focus on the results of the learning process and can consider the whole activity as very enjoyable, thus contributing to their elevated levels of happiness. In fact, learning a foreign language enables them to find a new purpose of their life, which reflects their inner need to study a foreign language (cf. [7]). This newly found purpose of life is again crucial as healthy aging is always positively correlated to purposefulness and meaningfulness.

In addition, the correlation between the demographic variables and the four factors confirmed other well-known facts about FLL. Firstly, although there were not any significant differences in the group of Polish seniors, the results among the Czech seniors indicated that younger seniors (55–65 years) appreciated the usefulness of FLL more than older ones. This is most likely connected with the fact that some of them still work and can use the foreign language in their jobs. Moreover, their economic situation enables them to travel and thus use a foreign language.

The findings also showed that the younger they were, they valued positive emotions and mental abilities while learning a foreign language more. The reason might be that younger seniors still feel more confident and less prejudiced about their FLL abilities [19]. This practical aspect of FLL can inevitably bring a feeling of the usefulness of one´s life even at an older age and therefore naturally increase the sense of meaningfulness.

Secondly, there were not any significant differences between the sexes in the assessment of the given four factors in both groups of respondents. Nevertheless, the Czech

male seniors seemed to be less worried about the negative aspects of FLL than the Czech senior females, and the Polish male seniors appeared to appreciate the positive emotions and mental abilities less than the Polish senior females. As research shows, this may be associated with the fact that females are more gifted than males in language learning and enjoy learning a foreign language more.

Furthermore, language is the most feminized field in public secondary education [20]. Although this might be considered a stereotype, it has been proven by evidence. For example, according to the survey conducted in 2013 in Britain, of all UK university languages students, 69% were female (19,775) and 31% were male (8935) [21]. As a Dutch study [22] also showed, female learners consistently outperformed male learners in speaking and writing. As this research is not intercultural but rather compares two regions that have a similar geographical area, the research does not bring any aspects of cognitive linguistics dealing with mental categories and the discrepancies between them when considering geographical differences. It only brings aspects of psycholinguistics, i.e., how a new cognitive activity, namely a second language acquisition at a later age, can contribute to increased levels of happiness and subjective well-being.

Thirdly, the correlation between the variable of education and the four factors in both groups of respondents revealed that people with a higher level of education were more likely to appreciate the benefits of FLL than those with a lower level of education. It is possibly connected to the better cognitive activity of the brain in those individuals with a higher intelligence quotient and also to the longer habitual activity, i.e., studying for a longer time in the past. Recollecting the moments from one´s life as a student can also contribute, at least unconsciously, to better feelings. These aspects would need further verification by subsequent research.

Finally, as far as the correlation between the variable of the number of languages studied and the four factors are concerned, the findings indicated that the more languages the senior knew, the subjectively happier and cognitively better, s/he felt. As it has been already pointed out by the theory of bilingualism, speaking two languages can bring not only cognitive benefits but also social benefits, e.g., communicating with other people of the same age and from different cultures, as well as psychological advantages (e.g., being less isolated).

The main limitation of this study consists of a relatively small scale of the research. This is caused by the fact that there are not very many courses for this age group and respondents from this age group. However, taking this into consideration, the research sample is sufficient and seems representative enough. Therefore, the research sample that may seem rather small, when taking into consideration this fact, in reality, is sufficient and enables us to draw statistically relevant conclusions. Another limitation might be that the research was conducted in two relatively small and neighbouring countries, but it can be an impetus for deeper and larger-scale research, ideally on a global level. Therefore, these limitations do not pose any threat to the relevance of this research and do not undermine its findings. Moreover, this research can be considered as a catalyst for further research activity into FLL and healthy aging.

In summary, all these aspects bring very important findings and can equip us with useful and efficient strategies, leading to better and healthy aging. As it has already been noted, older age brings many challenges and does not need to be viewed as a period of life that should rather be neglected or even eliminated [23]. Moreover, we have to attempt to meet the cognitive, mental and societal needs of older population groups, especially in this digital world, as it brings many challenges but also possibilities to improve or maintain healthy aging. FLL generally can be utilized as a tool for cognitive enhancement, not only in younger students [24–26], but also at an older age as our research clearly indicated.

## 5. Conclusions

The research conducted and presented in this paper summarizes the most important aspects of learning a foreign language in older age as a key strategy to maintain the

subjective feeling of happiness at a relatively high level in elderly people without necessary pharmacological intervention. This nonpharmacological intervention through increased cognitive activity and social contacts can very well supplement other tools widely used to enhance and sustain human well-being at an older age. The research clearly indicates that this approach can be valid as it brings many new opportunities to successful aging as a natural process and an inevitable part of human life. Foreign language learning as a second language acquisition at an older age is a cognitively stimulating and socially enhancing activity, undisputedly leading to an improved life and better aging. Moreover, it proves to be a systematic approach to improved life in older age and thus can provide us with individual resilience strategies leading to societal sustainability in the long run. However, further research is needed to verify all these findings and further develop future trajectory of research activity into healthy aging.

**Author Contributions:** Conceptualization, B.K. and M.P.; methodology, B.K. and M.P.; software, A.C.-E. and S.D.; validation, B.K. and M.P.; formal analysis, A.C.-E., S.D., B.K. and M.P.; investigation, B.K. and M.P.; resources, B.K. and M.P.; data curation, A.C.-E., S.D., B.K. and M.P.; writing—original draft preparation, B.K. and M.P.; writing—review and editing, B.K. and M.P.; visualization, A.C.-E., S.D., B.K. and M.P. All authors have read and agreed to the published version of the manuscript.

**Funding:** This research received no external funding.

**Institutional Review Board Statement:** All the respondents were students of the U3A at the University of Hradec Kralove and Wroclaw University of Economics and Business. The participants´ agreement with the intervention by the foreign language classes they attended was expressed by their involvement in those courses. All of the participants had signed the written consent with the language class intervention before the course itself. There was no experiment involving humans conducted, it was a regular language class. The data were collected by an anonymized questionnaire related to the participants´ satisfaction with the courses, those that are conducted after all classes at the end of each semester. For this reason, there was no need to have an agreement of an ethics committee of the university. No personal data were collected and the questionnaire submitted to them does not demand any legal measures required by the GDPR regulation of the EU.

**Informed Consent Statement:** All the participants provided their written consent before the language course had started.

**Data Availability Statement:** All the data are at disposal at the corresponding author upon request.

**Acknowledgments:** The paper is supported by the project entitled Excellence (2021) at the Faculty of Informatics and Management of the University of Hradec Kralove, Czech Republic.

**Conflicts of Interest:** The authors declare no conflict of interest.

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
