# Peer review of "Factor Analysis of Subjective Well-Being Sustainability through Foreign Language Learning in Healthy Older Individuals"

_sustainability, doi:10.3390/su13031590_

Round 1

Reviewer 1 Report

  1. Please illustrate your research framework with figures.
  2. Authors need to well- illustrate your viewpoint by relating your own methodology.
  3. Figures can help you to explain how the research work and provide international readership to get a better understanding of your academic work.
  4. Make it more clear, please. Purposes of the study? Research model?What model exactly?

     5.   Please reconfirm the Analysis of the Data

     6.   Update your references, please (from 2015-2020)

Suggested references:

1.English e-learning in the virtual classroom and the factors that influence ESL (English as a Second Language): Taiwanese citizens’ acceptance and use of the Modular Object-Oriented Dynamic Learning Environment

2. An Empirical Study of How the Learning Attitudes of College Students toward English E-Tutoring Websites Affect Site Sustainability

3. Designing a System for English Evaluation and Teaching Devices: A PZB and TAM Model Analysis

4.Application of Information Technology in Preschool Aesthetic Teaching from the Perspective of Sustainable Management

5.The optimal setting of A/B exam papers without item pools: A hybrid approach of IRT and BGP

6. Applying the UTAUT to Understand Factors Affecting the Use of English E-Learning Websites in Taiwan

7. Interactive Instructional Device

8. Teaching the right things and teaching the things right: Professional development for mariners

9.  Developing a System for English Evaluation and Teaching Device.

Author Response

Dear Reviewer,

Thank you very much for your useful, inspiring and pertinent suggestions, which have signiicantly contributed to the overall improvement of the whole article. Please see the attached file abour our responses, as well as the manuscript about our modifications that have been highlighted in yellow.

Best wishes,

Authors

Reviewer 2 Report

I found the paper to address an interesting topic. Even though the methods used in the paper are rather simple, I think that the value added to the scientific research by this paper resides in the analyses and discussions. I have the following observations:

  • Please better state the purpose of the paper in the introduction and separate the introduction from the literature review. Please present more in depth the papers in the field by discussing their methods and their results (you can do this by adding a separate section dedicated to the literature review);
  • Materials and methods section should present the period of time in which the questionnaire has been available, on which platforms and how the respondents knew about its existence in order to fill it. Personally, I found the sample to be quite small. 
  • Please add the questionnaire as an annex to the paper
  • Please bring evidence on why you have used a 6 points Likert scale rather than a 5-points Likert scale which is usually used in the literature. Please provide references for stating your point of view.
  • Please clearly state your hypothesis and discuss their validation (the last part of section 3 is rather unclear).

Author Response

(The authors gave the same response as above.)

Reviewer 3 Report

Well written paper overall. There are a couple of spots that need to be addressed before publication. 

Lines 301-302, for example, mention that the positive outcomes outweighed the negative one. The authors simply state that this is self-evident from the findings, but do not explain in detail how they arrive at that conclusion. Just walk the reader through the process of how you came to this conclusion a little more in detail and it will work very well. 

Line 362-373 talk about the limitations of the study. If you could talk about these limitations in more of a positive way that would be better. For example, rather than simply acknowledge the small sample size or that participants came from only 2 countries, emphasize a little more that this article is intended as a preliminary catalyst for future research. Emphasize how crucial its message is that there are effective, non-pharmaceutical means to improve aging and quality of life. Invite readers to use this research as a springboard to future investigations involving other populations. 

Author Response

(The authors gave the same response as above.)

Round 2

Reviewer 1 Report

Well done.

Reviewer 2 Report

Thank you for the revised version of the paper and for addressing the reviewers' comments. I have no further observations.